# Consistency of causal claims in observational studies: a review of papers published in a general medical journal

Camila Olarte Parra [1,2] Lorenzo Bertizzolo [2,3] Sara Schroter [4] Agnès Dechartres,[5] Els Goetghebeur[1]

[1]Applied Mathematics, Computer Science and Statistics, Ghent University, Gent, Belgium
[2]Clinical Epidemiology, Biostatistics and Bioinformatics, University of Amsterdam, Amsterdam, Noord-Holland, The Netherlands
[3]U 1153, Equipe Methods, INSERM, Paris, France
[4]Editorial, BMJ, London, UK
[5]Sorbonne Université, INSERM, Institut Pierre Louis d'Epidémiologie et de Santé Publique, AP-HP. Sorbonne Université, Hôpital Pitié Salpêtrière, Département de Santé Publique, F75013, Paris, France

**Correspondence to**
Dr Camila Olarte Parra;
Camila.OlarteParra@UGent.be

## ABSTRACT

**Objective** To evaluate the consistency of causal statements in observational studies published in *The BMJ*.

**Design** Review of observational studies published in a general medical journal.

**Data source** Cohort and other longitudinal studies describing an exposure-outcome relationship published in *The BMJ* in 2018. We also had access to the submitted papers and reviewer reports.

**Main outcome measures** Proportion of published research papers with 'inconsistent' use of causal language. Papers where language was consistently causal or non-causal were classified as 'consistently causal' or 'consistently not causal', respectively. For the 'inconsistent' papers, we then compared the published and submitted version.

**Results** Of 151 published research papers, 60 described eligible studies. Of these 60, we classified the causal language used as 'consistently causal' (48%), 'inconsistent' (20%) and 'consistently not causal'(32%). Eleven out of 12 (92%) of the 'inconsistent' papers were already inconsistent on submission. The inconsistencies found in both submitted and published versions were mainly due to mismatches between objectives and conclusions. One section might be carefully phrased in terms of association while the other presented causal language. When identifying only an association, some authors jumped to recommending acting on the findings as if motivated by the evidence presented.

**Conclusion** Further guidance is necessary for authors on what constitutes a causal statement and how to justify or discuss assumptions involved. Based on screening these papers, we provide a list of expressions beyond the obvious 'cause' word which may inspire a useful more comprehensive compendium on causal language.

## INTRODUCTION

Many researchers remain tempted to draw causal conclusions from observational data despite acknowledging that mere association is not causation because causal inference is the ultimate goal of most clinical and public health research.[1][2] Gold-standard answers are typically sought through randomised controlled trials (RCTs). The unique ability of RCTs to avoid confounding bias[3] has led to demands that empirical research must

## Strengths and limitations of this study

► By evaluating published observational studies in a general medical journal, we provided relevant examples of (ambiguous) causal statements.
► We focused on the abstract where clear messages are especially important, as many readers start by screening the abstract of the study.
► Comparing the submitted and published versions of the abstract allowed us to identify whether any causal claims were made or not as a result of the peer-review process.
► The focus on the use of causal language rather than the specific methods avoided discussion on the validity of underlying assumptions justifying causal inference in the setting studied.
► Assessing observational studies from a single journal allowed us to flag the inconsistent use of causal claims in this context, but not to estimate its prevalence more generally.

be drawn from randomised studies to justify causal statements.[4–6] RCTs are mainly used to assess the effect of a treatment or intervention but are not easily adapted to evaluate prognostic or risk factors rather than interventions.

There are however good reasons to look beyond RCTs for evidence on treatment effects. In many settings, RCTs are not feasible, ethical or timely and thus observational data are all that is available for some time, as in the recent COVID-19 crisis. Furthermore, observational studies typically involve broader real-world contexts than RCTs, where the costs and risks of experimentation suggest studying high-risk patients without major comorbidities.[7] This selection challenges generalisation to the target population. Highly selected populations with a usually short follow-up, render RCTs inappropriate to evaluate (long-term) unintended side effects. Trials further suffer from treatment non-compliance which complicates analysis, as treatment-specific populations lose the benefit of randomisation.

Recent International Council for Harmonisation of Technical Requirements for Pharmaceuticals for Human Use (ICH) E9 guidelines therefore emphasise the importance of causal estimands beyond intention-to-treat, such as per-protocol and as-treated analysis.[8 9]

Deliberately avoiding causal statements on a hoped-for causal answer brings ambiguity and contrived reporting.[10 11] Instead, authors should openly discuss the likely distance in meaning and magnitude between the data-based measure they are able to estimate and the desired targeted causal effect. Arguments would consider study design with additional assumptions in context.[12] Owing to decades of progress in statistical science (involving potential outcomes, directed acyclic graphs, propensity scores and more),[13] this allows for results, often unreachable by randomised trials, with a justified causal interpretation.[14]

In 2010, Cofield et al[5] assessed the use of causal language in observational studies in nutrition but deemed causal language inappropriate for all observational studies. In 2017, Adams et al[15] also considered that only RCTs allowed for causal inference, in their study assessing how people understand causal expressions in the news. From a different angle, Haber et al[16] examined whether the tone and strength of causal claims made in a given paper matched the language describing the findings in social media. Not surprisingly, they found stronger causal statements in the media in half of the cases, emphasising the importance of clear scientific messages.

To promote this, Lederer et al[17] recently published a guide for authors and editors on how to report causal studies in respiratory, sleep and critical care journals. Rather than circumventing the problem by asking to avoid causal language, they provide key elements that ensure valid causal claims.[18] Besides briefly explaining causal inference, they provide a definition of a confounder, outline how to identify confounding through so-called directed acyclic graphs and discuss how p values are often misinterpreted and how their value does not reflect the magnitude, direction or clinical importance of a given association. All these elements empower their target audience to critically assess observational studies.

To find out whether and how statements in study reports present confusing use of causal language (or lack thereof), we examined research papers concerned with exposures and outcomes published in *The BMJ* in 2018. Our focus was on the causal message *The BMJ* readers receive from these papers, particularly from the abstract. We evaluate the consistency of causal statements in the published abstracts of observational studies, whether this consistency was a reflection of the full text and if any a priori changes had been made as a result of the peer-review process.

## METHODS
### Sampling and inclusion criteria
COP identified all original research articles published in *The BMJ* in 2018 described as either cohort or longitudinal studies in the study design section of the abstract. The eligible studies were identified by statements in this section of the abstract such as 'cohort', 'longitudinal' or 'registry-based'. Those identified as 'observational' were included if they suggested a period of follow-up rather than being cross-sectional. Articles described as case cohorts were excluded as their interpretation and analysis differs from other studies with follow-up assessing the exposure-outcome relationship.

### Assessment of published abstracts
Two reviewers (COP, LB) independently screened the published abstracts of the eligible papers. For the text included under each of the subheadings in the abstract (objective, design, setting, participants, outcome, results, conclusion), the reviewers assessed whether there was an (implicit) causal (cl)aim using a yes/no/unclear response. After assessing each separate subheading, each reviewer then gave an overall assessment of the main claims in the paper's abstract as either 'consistently causal', 'inconsistent' or 'consistently not causal'. After the independent assessments, the overall rating of the abstract was compared between both reviewers; where there was disagreement, a third reviewer (EG) was consulted and a consensus reached.

### Assessment of published full text
We further evaluated the full published text of all eligible papers to identify the statistical methods applied and any further causal claims. In particular, we looked for statements that would support or undermine a causal aim, including confounding adjustment, discussing residual confounding, exchangeability and issues of transportability. We randomly divided the papers between the two reviewers (COP, LB) for this assessment. For each paper, we extracted statements where authors described the statistical method and method for confounding adjustment, if any. We then extracted the sentences summarising the results and conclusions to highlight any causal claims.

### Assessment of initially submitted abstract version
As the focus of this paper is to highlight ambiguous use of causal language, we further assessed those articles judged as 'inconsistent' to see if there were changes introduced to the manuscript between submission and publication, leading to this inconsistent use of causal language. For this subset, we obtained the submitted version of the manuscripts and the associated peer reviewers' comments from *The BMJ*'s manuscript tracking system. We then compared the published version with the first submitted version of the abstract to identify whether the same wording related to causal claims appeared in the submitted version and whether changes occurred as a result of comments from peer reviewers and editors, as indicated in the corresponding peer-review reports.

The same reviewers (COP, LB) independently evaluated the submitted versions of the abstracts. The reviewers assessed whether the content under each subheading of

the submitted abstract differed from the published version. Where there were discrepancies between versions, each reviewer indicated the presence of a causal claim as yes/no/unclear for each abstract subheading (title, objective, design, setting, participants, outcome, results, conclusion) and made an overall assessment of the submitted abstract as either 'consistently causal', 'inconsistent' or 'consistently not causal'. As before, the assessments were compared and, in cases of disagreement, a third reviewer (EG) was consulted and consensus reached.

### Patient and public involvement

Patients were not involved in the design, analysis or interpretation of the study. Patients were not participants in this study; it was a methodological study (research on research). Patients' opinions of causal statements and the use of ambiguous language in research papers is important and further work in this area partnered by patients is important.

### RESULTS
### Assessment of published abstracts

In 2018, 151 research papers were published in *The BMJ*, of which 60 (40%) were eligible for inclusion in our study. We identified 29 studies (48%) reporting causal language consistently. A further 12 (20%) studies were considered inconsistent mainly because the objective stated the evaluation of an association while the conclusion presented a causal finding (9/12) or the opposite (3/12). Finally, there were papers that described studies aiming for prediction or reporting associations without (implicitly) suggesting that they had a causal nature that were considered consistently not causal (n=19, 32%). Table 1 shows sample excerpts from the published abstracts that were evaluated. Each row corresponds to statements from the same study. The first column indicates the assigned category, based on the type of association it describes. The last column explains why a given abstract was considered to belong to the assigned category. As the assessment pertains to causal claims in general, the words referring to the particular topic of the corresponding study were removed from the statements. The examples shown are not an exhaustive list, but were chosen to illustrate the different phrasing of statements belonging to the different categories. It is worth noting that the statements presented correspond to the objective and conclusion subheadings of the abstract. When assessing the abstracts, we identified that these were the subheadings under which the information to classify the abstract was mainly found. Other subheadings like design, setting and participants were not as relevant for this purpose, but were also assessed.

To further illustrate how statements in these two sections can be misleading, we tabulated a few examples in a 2 by 2 table showing mismatches between what was reported in the objectives and conclusion resulting in

the paper being categorised as either 'consistently (not) causal' or 'inconsistent' (table 2).

### Assessment of published full text

Table A in online supplemental material presents statements found in both the published abstract and published full text of each of these papers (n=60) regarding the statistical method used and considerations suggesting a causal aim or otherwise. Each row corresponds to a different study. The papers are grouped according to the category to which the corresponding abstract was assigned to. The particular causal or non-causal wording is highlighted in bold. A brief description on the consistency of causal language is provided in the last column of table, labelled 'Comment'.

We found that all papers classified as 'consistently causal' based on the abstract, also used causal language and contained causal statements in the full text. This was additionally the case with more than half (11/19) of the abstracts classified as 'consistently not causal', where even though the abstract was carefully phrased in terms of association, the authors applied causal methods, discussed residual confounding, biological plausibility or a dose-response relationship suggesting a causal aim.

In the previous section, we referred to three abstracts that had a clear causal objective but a non-causal conclusion. In the full text of these papers, the authors discussed concerns of residual confounding which explains why they decided to play down the conclusion.

Looking at the 'Methods' section in the full text of the abstracts classified as 'inconsistent', we found that 11 of the 12 provided adjusted estimates. Most of the studies (8/12, 67%) used outcome regression models, mainly Cox proportional hazard models, or (propensity score) matching (3/12, 25%).

### Assessment of submitted abstract version

Of the 12 published abstracts classified as 'inconsistent', we further classified 11/12 (92%) as also inconsistent on submission. There was only one study where the submitted version of the abstract described a different type of association. In this case, the conclusion of both the submitted and published versions was rather conservative by stating that the intervention was 'independently associated' with the outcome. The submitted version expressed a causal objective, stating the aim of evaluating the 'impact' of a particular intervention with corresponding methods: providing adjusted estimated effects and including sensitivity analysis using propensity score matching. However, in the published version the term 'impact' was replaced by 'association' making the abstract less clear about a causal aim because both the abstract's objectives and the conclusion described an association but the authors still provided adjusted HRs and resorted to propensity score matching.

### DISCUSSION
### Statement of principal findings

We found that the majority (80%) of the published research abstracts reporting on observational studies had

**Table 1**  Examples of statements found in the objectives and conclusions sections of abstracts of observational studies published in *The BMJ* in 2018 and their corresponding assigned category

| Assigned category | Abstract objectives | Abstract conclusions | Comment |
|---|---|---|---|
| Consistently causal | '…assess the effectiveness of…' | 'Little evidence was found of a direct impact of…' | When discussing associations, words like effect, contribution or role are similar to cause and then (direct) impact and effect will be their consequence. |
| | 'To determine the effect of … in …' | '…has led to risk…' | |
| | 'To describe the contributions of…' | '… an important role in …' | |
| | 'To evaluate the impact of …' | '…impacts are…' | |
| | 'To investigate whether improving adherence to …' | '…the beneficial effect of improved…' | Evaluates taking an action 'improving adherence' and concludes that the effect is beneficial. |
| | '…benefit of … in reducing … risk' | '… is an overlooked risk factor for …' | Evaluates how a given intervention can reduce the risk of an outcome and then labels it as an 'overlooked risk factor'. |
| | 'To determine outcomes and safety of…' | '… is at least as effective and safe as …' | Evaluates and determines that a certain intervention is as safe as the comparator. |
| | 'to quantitatively decompose this joint association to … only, to … only, and to their interaction'. | '…excess risk of…These findings suggest that most cases of … could be prevented by …' | Suggests interest in direct and indirect effect, that is, mediation analysis, and concludes consequently. |
| Consistently not causal (associations) | '…is associated with …compared with…' | '…is associated with …compared with…' | Describes associations without labelling them as causal or prediction. |
| | 'To describe trends in…' | '…rates were high during the study period of … with the highest rates in … vs …' | Limits to describe frequency. |
| | 'To assess how often …' | 'One in … adults … were …' | |
| | 'To examine the association between…' | '…could increase … confirmation of these findings are warranted, preferably in an intervention setting'. | Suggests further research to determine the nature of the association. |
| | '…compared with…is associated with…' | 'Additional studies, with long term follow-up, are needed to investigate the effects of…' | |
| Consistently not causal (prediction) | 'To develop and validate a set of practical prediction tools that reliably estimate the outcome of…' | '…prediction models reliably estimate the outcome…' | Describes developing and validating prediction models. |
| | 'To prospectively validate the … algorithm to …' | '…accurately classified…' | |
| Inconsistent | '…evaluate safety of…' | '…associated with…' | Phrasing the objective as causal and limiting to describing an association in the conclusion. |
| | '…analyse the effect of…' | | |
| | '…critical determinant…' | | |
| | '…association with…' | '…is safe…' | Phrasing the objective as just to explore an association and presenting a causal claim in the conclusion. |
| | | '… had no substantial effect on long term survival…' | |
| | | '… was determined by… may be largely explained by…' | |
| | | '… was found to be the safest drug, with reduced risks of…' | |
| | | 'These results emphasise the benefit of…' | |
| | '…association with…' | '…tackling all these risk factors might substantially…' | Phrasing the objective and conclusion as if just to assess an association but then suggesting to take action given the findings. |
| | | '…Targeting … prevention strategies among these patients should be considered'. | |
| | | 'Systematically addressing … may be an important public health strategy to reduce the incidence of' | |
| | | '…present findings encourage the downward revision of such guidelines …' | |

a consistent use of causal language in the abstract. Still 20% of abstracts contained inconsistent messages on the causal nature of the key 'effect'. Inconsistencies showed up in two directions: an intentional quest for causality ending in uncriticised non-causal conclusions or carefully phrased mere associations ending with recommendations to act and intervene based on the exposure outcome association.

Beyond the wording, readers can learn much about the sought, after interpretation from described statistical methods, and assumptions made explicit in the paper. On a case-by-case basis, one could then assess whether additional assumptions, for example, involving 'no-unmeasured confounders', would justify the causal assessment derived from these approaches. Identifying key elements like the ones presented in the online supplemental

**Table 2** Examples of (mis)matching causal and non-casual statements found in the objectives and conclusions sections of abstracts of observational studies published in *The BMJ* in 2018

| | | Abstract conclusions | |
| --- | --- | --- | --- |
| | | **Causal** | **Not causal** |
| Abstract objectives | Causal | Consistent<br>'…assess the effectiveness of…'and 'Little evidence was found of a direct impact of…'<br>'…benefit of … in reducing … risk' and '… is an overlooked risk factor for …' | Inconsistent<br>'…evaluate safety of…' and '…associated with…'<br>'…analyse the effect of…' and '…associated with…'<br>'…critical determinant…' and '…associated with…' |
| | Not causal | Inconsistent<br>'…association with…' and '…is safe…'<br>'…association with…' and '… had no substantial effect on long term survival…'<br>'…association with…' and '…tackling all these risk factors might substantially… '<br>'…association with…' and 'Systematically addressing … may be an important public health strategy to reduce the incidence of' | Consistent<br>'To describe trends in…' and '…rates were high during the study period of … with the highest rates in … vs …'<br>'To assess how often …' and 'One in … adults … were …'<br>'To develop and validate a set of practical prediction tools that reliably estimate the outcome of…' and '…prediction models reliably estimate the outcome…' |

material would help to assess if causal inference is possible. If in doubt, a sensitivity analysis may be in order. It seems better to be transparent about the ultimate aim to draw a causal conclusion and to acknowledge to fall short of that, than to generate confusion.

When assessing the full text of the 'consistently causal' papers, we identified that authors often discussed these assumptions and resorted to conducting a sensitivity analysis. This was also the case for those papers that were classified as 'inconsistent' or 'consistently not causal'. In these papers, there was a concern for residual confounding because of the observational nature of the study or due to specific missed confounders. Therefore, the abstract's objective avoided suggesting a causal aim instead of being explicit of such concern or limitations in the abstract.

### Comparison with other studies

This is not the first study to evaluate the use of causal language in the medical literature. Cofield *et al*[5] assessed the use of causal language in observational studies in nutrition. However, they focus only on assessing whether authors included causal language or not, as it was deemed inappropriate due to the observational nature of the study. We have made the case that merely avoiding explicit causal terms is not a real solution. Even without them, a causal conclusion is implicit when the take home message encourages interventions based on the presented findings. Avoiding inconsistency is important but equally one

should be able to trust that the use of consistent causal language is not in vain. This requires a more in-depth look at methods and assumptions validating the causal claims.

### Strengths and weaknesses of the study

Accurate abstracts are important. In just a few brief paragraphs, the authors summarise key elements of design, methods and results, and come to a conclusion. Many readers only read the abstract. However, a powerful abstract opens the door to readers and sets the scene for any study. It serves the different roles of informing the audience about its main findings while motivating the reader to further explore the full text, all within the constraints of brevity. This demands authors to give special attention to ensure that every word in the abstract is required. All of the above makes the assessment of the abstract relevant but also challenging.

Further research is needed to explore how causal claims presented in the abstract and full text are supported by the design and methods applied, which entails assessing the methods used and evaluating whether the underlying assumptions were met.[19] The optimal conclusion should not simply label a study as black or white in causal terms. In the present study, we used a convenient limited number of classifications for short statements. In practice, a continuous degree of confidence in a potential causal relationship is likely to emerge based on the observed association.

**Table 3** Impact of the errors of causal effect assignment

| | | True nature of the main exposure effect | |
| --- | --- | --- | --- |
| | | **Causal** | **Not causal** |
| Reported nature of the studied exposure effect | Causal | A true causal effect has been discovered. Recommendation to act on this should be considered. Language in the context of a study intended for causal inference. | Type I error: there is no causal effect, but it is claimed. Causal language used or suggestion to take action made when the purpose/ability was to find associations. |
| | Not causal | Type II error: hiding the true causal objective/result by avoiding use of causal language. | No causal language when the objective is prediction or to explore associations. |

**Table 4** Examples of words and study elements that could point to causality or otherwise

| | |
|---|---|
| Words expressing a causal relationship | ► Affect<br>► Attributable<br>► Benefit<br>► Cause/Causal pathway<br>► Contribute<br>► Determinant<br>► Effect<br>► Efficacy<br>► Impact<br>► Improve<br>► Leads to<br>► Mediates<br>► Responsible for<br>► Results in<br>► Safety |
| Words that could suggest causality in a given context | ► Independently associated<br>► Induce<br>► Higher (lower) probability<br>► Modify<br>► Risk (factor)<br>► Trajectory (quantitatively) decompose |
| Specific expressions avoiding suggestions of causal effects | ► Association<br>► Correlation<br>► Less (more) likely link<br>► Predict<br>► Pattern |
| Key aspects suggesting causal aim | ► Adjusting for confounders<br>► Discussing biological plausibility, dose-response and/or temporal relationship<br>► Discussing 'unmeasured confounders' assumption<br>► Mediation analysis<br>► Propensity score adjustment (propensity score) matching<br>► Providing estimates of (population) attributable risks<br>► Suggesting/Recommending intervention<br>► Target trial emulation design<br>► Using directed acyclic graphs to identify confounders and mediators<br>► Using negative controls<br>► Using instrumental variables |

We are aware that by limiting our assessment to the consistency of causal language, we may have missed the discussion of the extent to which the underlying assumptions that enable causal inference were met. This requires subject-matter knowledge in each particular case.[13] Indeed, when there was a clear causal aim but the authors considered that these assumptions were not fulfilled, they may have decided that a causal claim was inappropriate and phrased their conclusion in terms of association rather than causation. If this is the case, the apparent inconsistency would no longer hold. On the contrary, any undue causal claims can be viewed as a form of spin.[20 21]

## Policy implications

As observational data resources abound, methods for causal inference from observational data have surged in tandem with the call for real-world evidence. The new opportunities bring new challenges and the responsibility for clear and well-supported statements on the evidence. In this spirit and motivated by novel guidelines as proposed by International Council for Harmonisation of Technical Requirements for Pharmaceuticals for Human Use (ICH) E9 and Food and Drug Administration, Hernán *et al* have embarked on a project entitled 'Developing Guidelines for the Analysis of Randomised Controlled Trials in Real-World Settings'.[22] The importance of such initiatives supports a shift towards being explicit and discussing assumptions underlying causal methods that allow for causal interpretations in context, with or without an RCT.[13] In the meantime, uncritical ambiguous phrasing in observational studies remains prevalent.[14] Those searching for the best possible evidence supporting future treatment decisions are best served by transparent reports of observational studies.

Faced with uncertainty when concluding on the nature of the observed exposure outcome relationship, a justifiable balance between the type I and II error rate is a natural guide for action. The cost of errors must be weighed in context, for instance, as in clinical trials emphasising control of the type I error to avoid introducing new unhelpful drugs at a potentially large cost. Alternative weights are typical in screening programmes where false positives will be caught in follow-up examinations, but false negatives are lost forever. In a crisis, such as the current COVID-19 pandemic, we must act before long-term randomised trials have materialised. It becomes undeniably important to learn as much as we can from observational data, be aware of the types of risk when acting or not, as displayed in table 3.

A prerequisite for good causal language practice includes awareness of which language implies a causal statement and which does not. To support correct phrasing and raise awareness, we have compiled a short list of words and expressions with dedicated (non) causal meaning (table 4). The list draws on phrases found in our study and in the references cited, particularly Hernán[10] and Thapa *et al*.[6] This list is a suggestion as a starting point and further studies can test and validate it. We consider that a definition of causal language that is generally recognised by the research community is needed.[23 24]

Words like 'effect', 'impact', 'determinant of'…, inevitably point in the causal direction and their use should come with the requirement of at least stating and ideally critically evaluating the necessary assumptions.[6] Uncertainty on the causal nature of the conclusion should tone down any suggestion for intervening on the studied exposure. Specifying the corresponding level of evidence rather than hiding the ultimate causal aim of a study is what we recommend,[19] while acknowledging a margin of error in any empirical study.[20]

## Conclusion

In summary, we have found that causal messages are embedded in studies otherwise carefully phrased in terms of association. Further guidance for authors is needed on

what constitutes a causal statement, similar to the one published by Lederer et al[17] for respiratory, sleep and critical care journals. We look forward to similar guidance for other disease groups. From the screened BMJ abstracts, we provided a list of expressions with clear interpretation which may inspire a useful more comprehensive compendium that can be derived from a consensus meeting, for instance. We argue that such awareness and special attention among authors and reviewers would serve our communication on the best available evidence for conceived interventions.

**Correction notice** The affiliation for Agnès Dechartres has been corrected to Sorbonne Université, INSERM, Institut Pierre Louis d'Epidémiologie et de Santé Publique, AP-HP. Sorbonne Université, Hôpital Pitié Salpêtrière, Département de Santé Publique, F75013, Paris, France

**Contributors** Conceptualisation: COP, LB, AD, SS, EG. Methodology: COP, EG. Data curation: COP, LB. Formal analysis: COP, LB, EG. Writing original draft: COP. Writing—review and editing: LB, SS, AD, EG. Approved final version: COP, LB, AD, SS, EG.

**Funding** This project has received funding from the European Union's Horizon 2020 research and innovation programme under the Marie Sklodowska-Curie grant agreement no. 676207.

**Competing interests** SS is a full-time employee of the BMJ Publishing Group. No other competing risks to declare.

**Patient consent for publication** Not required.

**Ethics approval** We did not seek ethical approval or consent from authors and reviewers for this descriptive quality improvement study. However, when authors and reviewers submit manuscripts and reviews to *The BMJ*, they are notified that their paper or review may be entered into research projects for quality improvement purposes. COP was given access to *The BMJ*'s data under a confidentiality agreement. We do not report any identifying information. The statements presented in this study belong to published papers available in the public domain.

**Provenance and peer review** Not commissioned; externally peer reviewed.

**Data availability statement** All data relevant to the study are included in the article or uploaded as supplementary information. The reviews and published versions of the papers included in the study are publicly available at BMJ.com. No further data will be made available as it is confidential submission data.

**ORCID iDs**
Camila Olarte Parra http://orcid.org/0000-0003-0263-4392
Lorenzo Bertizzolo http://orcid.org/0000-0002-1666-2450
Sara Schroter http://orcid.org/0000-0002-8791-8564

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
