## [Reviewer comments · BMJ Open]

ARTICLE DETAILS

TITLE (PROVISIONAL)	Consistency of causal claims in observational studies: a review of papers published in a top general medical journal
AUTHORS	Olarte Parra, Camila; Bertizzolo, Lorenzo; Schroter, Sara; Dechartres, Agnes; Goetghebeur, Els

VERSION 1 – REVIEW

REVIEWER	Lex Bouter department of Epidemiology and Data Science Amsterdam University Medical Centers The Netherlands
REVIEW RETURNED	20-Aug-2020

GENERAL COMMENTS	This is an interesting analysis of the consistency of causal statements in the abstracts of 60 BMJ articles that report on observational studies of treatment effects. It's well written but has a number of major issues that need attention. Major issues  [ ] Table 4 illustrates my main concern. I fail to understand why only for the 12 articles with inconsistent causal claims the full text was studied to assess whether a causal interpretation was justified. Obviously that needs to be done for all 60 included articles. The reason is that – as table 4 explains – also consistent causal claims and consistent non-causal claim can be wrong in the sense that the full text doesn't support them. So in my view table 4 needs to have three rows (causal, inconsistent, not causal claim in the abstract) and two columns (true nature of the main effect is causal or not causal on the basis of the full text. With all 60 articles allocated to one of these 6 cells this would be the core table of the manuscript. [ ] None of the essentials of data extraction is clearly operationalized. How exactly was it decided whether the causal statement in the abstract was inconsistent, consistent causal (including suggest causal because that seems to be the same thing), or consistent non-causal? How was it decided whether the full text justifies a causal claim or not? The assessments were done by two persons independently, but what they exactly did and what their level of agreement was is not presented. [ ] There is no link to a preregistered study protocol or a registered report. Also I believe that the data extraction results database can and should be made available in anonymized form. [ ] The authors fail to explain that they study an aspect of 'quality of reporting'. More specifically the reader would like to know if the authors believe that STROBE covers the quality of reporting of causal claims well enough or whether there should be a STROBE extension or modification for this topic. Minor issues  [ ] The authors adequately explain the reasons why there is – next to
---

	RCTs – a place for observational intervention research but miss one important reason. RCTs are notoriously unsuitable for studying serious unintended side effects of interventions due to their selectivity and relatively small number of participants.  □ It's unclear whether the 60 out of 151 articles were selected by two persons independently as well. □ I'm not fully convinced that no approval by an Ethics Committee would have been necessary for this study. The manuscript only says that corresponding authors were informed that their manuscript and its reviews might be used for research. That seems alright but is not ethics approval. □ The term 'effect' seems to belong to the upper part of the Box as it's clearly expressing causality. □ Table 3 is very lengthy and not essential to follow the reasoning. Would be more suitable as digital supplement.
--	--

REVIEWER	Regina Nuzzo Gallaudet University, USA
REVIEW RETURNED	03-Sep-2020

GENERAL COMMENTS	This is an interesting study on the use of causal statements in observational study abstracts published in The BMJ in 2018, particularly in the discrepancies between stated objectives and conclusions. This area of study is important, and the authors had access to a rich and novel set of data. Below I discuss questions and suggestions for the authors. The authors state that their objective was to evaluate the “consistency of causal statements” in the studied abstracts, and in the methods they write that since “the focus of this paper is the avoidance of misleading and ambiguous messages, we further assess those articles judged as ‘inconsistent’ [with respect to causal objectives and conclusions in the abstract] to see if there were changes introduced to the manuscript between submission and publication.” It's not clear to me why only inconsistently causal abstracts were selected for further comparison with submissions and full-text assessments. It would be perhaps even more interesting to learn about the submissions that led to ‘consistently not causal’ abstracts (eg, were previously causal statements weakened during peer review?) or the full text behind ‘consistency causal’ abstracts (eg, did their statistical methods align with Lederer et al (2019)’s guidelines?). A focus on “the avoidance of misleading” messages seems to point to an analysis of those abstracts most likely to contain misleading claims, which would be those that made consistently causal statements. I would also have liked more explanation about what criteria the two reviewers used to make the determination of the presence of a causal claim. The tables provide examples after the fact, and the box provides suggestions, but a more rule-based, systematic discussion would be helpful to better assess the authors’ methods here. For example, I'm curious how the authors decided to classify statements that suggest taking action given the findings; are these strictly causal statements? Perhaps most importantly, the ultimate goal of this study wasn't entirely clear to me. Do the authors hope to document the confusion that seems to exist for authors and peer reviewers around causal language, with the hopes that journals such as The BMJ will provide guidelines for best practices in causal language (with resources
--

	such as the Box provided in this paper)? If so, it seems that it would be important to study all the abstracts of observational studies in the sample (not just the 'inconsistent' ones) to document how submitted causal/non-causal language is changed or not during peer review. In this case, the assessment of statistical methods in the full text is not very important. On the other hand, if the authors hope to document use the use of causal language in published abstracts and explore how misleading and ambiguous this language is, then examining the full text of all papers with any kind of causal statement in the abstract would seem to be most appropriate. Overall, I enjoyed reading this manuscript, and I appreciated the opportunity to review it.
--	---

REVIEWER	Luke Bratton United Kingdom
REVIEW RETURNED	09-Sep-2020

GENERAL COMMENTS	Investigating the consistency of language in journal articles is an important avenue of inquiry. There is a tendency for the blame to be placed on news reporters, or press officers, when a scientific headline contains an inaccuracy – a source of misinformation for the public. Scientists have to take into consideration that inconsistent language in their own research would be a source of misinformation for other scientists and future publications, but also a source of misinformation to press officers, news reporters, and the public. This paper should prompt others to consider the inconsistencies in language in research papers in their own fields, and should prompt a deeper analysis of the origins and solutions for such inconsistencies. In general this paper is well written, structured, and formulated. I only have a few very minor suggestions for your consideration below. Page 4 Line 50 Page 8 Line 215 Typo: "ICH E9" guidelines, rather than "ICH9". Page 5 Line 82 States "statements [...] such as "cohort", "longitudinal" or "registry-based"". This suggests this is not an exhaustive list for selection of eligible studies. If studies were selected based on other criteria, what are these? Would an individual attempting a replication make the same selections based on the information you have provided here? Page 9 Line 235-246 For this discussion of the classification of statements of causality, I would point you to research by Adams et al (2017) regarding how such statements are interpreted and classified by participants. Any such classification system for causality statements might need to be produced by widespread consensus rather than, or in addition to, classifications created by experts. I add this point as a suggestion because the author's final statement on the matter is that there should be "further guidance", but not what that further guidance should be, or how the guidance should be informed.
--

VERSION 1 – AUTHOR RESPONSE

Reviewer # 1: Lex Bouter

This is an interesting analysis of the consistency of causal statements in the abstracts of 60 BMJ articles that report on observational studies of treatment effects. It's well written but has a number of major issues that need attention.

Major issues

- Table 4 illustrates my main concern. I fail to understand why only for the 12 articles with inconsistent causal claims the full text was studied to assess whether a causal interpretation was justified. Obviously that needs to be done for all 60 included articles. The reason is that – as table 4 explains – also consistent causal claims and consistent non-causal claim can be wrong in the sense that the full text doesn't support them. So in my view table 4 needs to have three rows (causal, inconsistent, not causal claim in the abstract) and two columns (true nature of the main effect is causal or not causal on the basis of the full text. With all 60 articles allocated to one of these 6 cells this would be the core table of the manuscript.

Following this suggestion and a similar suggestion from reviewer # 2, we have assessed the full published text of all the included papers (n=60). The corresponding statements extracted for all the papers are presented in Table A of the Supplementary Material. Reviewing the full text of all the included studies allowed us to extend Box 2, where we enumerated the terms and study elements that point to causality or otherwise. We were also able to contrast whether the consistency of causal language, as assessed in the abstract, was a reflection of the main text. To further detail this extraction process and the corresponding results, we included a new section in Methods (lines 97-104) and Results (lines 159-173).

- None of the essentials of data extraction is clearly operationalized. How exactly was it decided whether the causal statement in the abstract was inconsistent, consistent causal (including suggest causal because that seems to be the same thing), or consistent no-causal? How was it decided whether the full text justifies a causal claim or not? The assessments were done by two persons independently, but what they exactly did and what their level of agreement was is not presented.

As explained, in the previous point, Table A in supplementary material presents result statements of all the evaluated papers. We included a column with a comment on how these statements helped classify each of the papers in the different categories. To avoid confusion, we decided to combine the categories 'consistently causal' and 'suggest causal' under the label 'consistently causal'.

There is no link to a preregistered study protocol or a registered report. Also I believe that the data extraction results database can and should be made available in anonymized form.

We have now included the data extraction in an anonymized form as Supplementary Material. We did not provide a link to a preregistered study protocol because the study protocol was not registered in advance.

The authors fail to explain that they study an aspect of 'quality of reporting'. More specifically the reader would like to know if the authors believe that STROBE covers the quality of reporting of causal claims well enough or whether there should be a STROBE extension or modification for this topic.

The aim of our paper is to flag the issue of the inconsistent use of causal language in observational studies and provide specific examples. We consider that authors would benefit from further guidance.

This also includes further research on how to ensure that the consistency of causal language is enhanced with the STROBE checklist.

Minor issues

□ The authors adequately explain the reasons why there is – next to RCTs – a place for observational intervention research but miss one important reason. RCTs are notoriously unsuitable for studying serious unintended side effects of interventions due to their selectivity and relatively small number of participants.

This is indeed an important limitation of RCTs and we have included a corresponding statement in the introduction (lines 47-48).

□ It's unclear whether the 60 out of 151 articles were selected by two persons independently as well.

As stated in the Method section, this selection was not performed in duplicate (line 82-83). One reviewer (COP) selected the studies and then two reviewers (COP and LB) independently assessed the abstract to classify them according to the consistency of causal language. The format of The BMJ facilitated the selection of the papers because they require that the study design is stated in the title and also the structured abstract includes a study design section. Therefore it was possible to easily identify the cohort studies from the information provided in these sections.

□ I'm not fully convinced that no approval by an Ethics Committee would have been necessary for this study. The manuscript only says that corresponding authors were informed that their manuscript and its reviews might be used for research. That seems alright but is not ethics approval.

We consider that this study did not require ethical approval for several reasons. First, it did not involve human subjects. Second, the submitted manuscripts, published versions and reviewers comments are publicly available in The BMJ website. Finally, we do not report any identifying information, besides what is already in the public domain.

□ The term 'effect' seems to belong to the upper part of the Box as it's clearly expressing causality.

Indeed, the word 'effect' indicates causality and has been moved to the corresponding section.

□ Table 3 is very lengthy and not essential to follow the reasoning. Would be more suitable as digital supplement.

Table 3 is now part of the Supplementary material as Table A. As noted before, it has been modified to include statements for all the papers, not only from the papers labelled 'inconsistent'.

Reviewer # 2: Regina Nuzzo

This is an interesting study on the use of causal statements in observational study abstracts published in The BMJ in 2018, particularly in the discrepancies between stated objectives and conclusions. This area of study is important, and the authors had access to a rich and novel set of data. Below I discuss questions and suggestions for the authors.

The authors state that their objective was to evaluate the “consistency of causal statements” in the studied abstracts, and in the methods they write that since “the focus of this paper is the avoidance of misleading and ambiguous messages, we further assess those articles judged as ‘inconsistent’ [with respect to causal objectives and conclusions in the abstract] to see if there were changes introduced to the manuscript between submission and publication.” It’s not clear to me why only inconsistently causal abstracts were selected for further comparison with submissions and full-text assessments. It would be perhaps even more interesting to learn about the submissions that led to ‘consistently not causal’ abstracts (eg, were previously causal statements weakened during peer review?) or the full text behind ‘consistency causal’ abstracts (eg, did their statistical methods align with Lederer et al (2019)’s guidelines?). A focus on “the avoidance of misleading” messages seems to point to an analysis of those abstracts most likely to contain misleading claims, which would be those that made consistently causal statements.

As we explained above in response to reviewer # 1, we have now assessed the full published text of all the included studies (n=60). The data extraction is presented in Table A of the Supplementary material. When evaluating the full text, we were able to contrast whether the consistency of causal language, as assessed in the abstract, was a reflection of the main text. After reviewing the full text of all the included studies, we also extended Box 2, where we enumerated the terms and study elements that point to causality or otherwise. To further detail this extraction process and the corresponding results, we included a new section in Methods (lines 97-104) and Results (lines 159-173). We also rephrase the quoted statement (lines 106-108).

I would also have liked more explanation about what criteria the two reviewers used to make the determination of the presence of a causal claim. The tables provide examples after the fact, and the box provides suggestions, but a more rule-based, systematic discussion would be helpful to better assess the authors’ methods here. For example, I’m curious how the authors decided to classify statements that suggest taking action given the findings; are these strictly causal statements?

We expect that the explanations included under the ‘Comment’ column in Table A of the Supplementary material, help clarify how the consistency in the use of causal language was evaluated and whether the abstract reflected the corresponding consistency of the full text.

Perhaps most importantly, the ultimate goal of this study wasn’t entirely clear to me. Do the authors hope to document the confusion that seems to exist for authors and peer reviewers around causal language, with the hopes that journals such as The BMJ will provide guidelines for best practices in causal language (with resources such as the Box provided in this paper)? If so, it seems that it would be important to study all the abstracts of observational studies in the sample (not just the ‘inconsistent’ ones) to document how submitted causal/non-causal language is changed or not during peer review. In this case, the assessment of statistical methods in the full text is not very important. On the other hand, if the authors hope to document use the use of causal language in published abstracts and explore how misleading and ambiguous this language is, then examining the full text of all papers with any kind of causal statement in the abstract would seem to be most appropriate.

We consider that by providing concrete and real examples, we can open the debate about the use of causal language in observational studies. Our goal is to give specific examples of how these statements can be misleading, as detailed in the Supplementary Material. We consider that the elements listed in Box 2, are a starting point to provide further guidance for authors and peer reviewers. But, as we explained in the discussion, this is by no means an exhaustive list and further research is needed to provide a more comprehensive compendium. As we clarify in the discussion, further research is needed to evaluate how the choice and application of the (statistical) methods support the causal claims and allow for a causal claim to be valid.

Overall, I enjoyed reading this manuscript, and I appreciated the opportunity to review it.

Reviewer # 3: Luke Bratton

Investigating the consistency of language in journal articles is an important avenue of inquiry. There is a tendency for the blame to be placed on news reporters, or press officers, when a scientific headline contains an inaccuracy – a source of misinformation for the public.

Scientists have to take into consideration that inconsistent language in their own research would be a source of misinformation for other scientists and future publications, but also a source of misinformation to press officers, news reporters, and the public.

This paper should prompt others to consider the inconsistencies in language in research papers in their own fields, and should prompt a deeper analysis of the origins and solutions for such inconsistencies.

In general this paper is well written, structured, and formulated. I only have a few very minor suggestions for your consideration below.

Page 4 Line 50

Page 8 Line 215

Typo: "ICH E9" guidelines, rather than "ICH9".

The typo has been corrected accordingly.

Page 5 Line 82

States "statements [...] such as "cohort", "longitudinal" or "registry-based"".

This suggests this is not an exhaustive list for selection of eligible studies. If studies were selected based on other criteria, what are these? Would an individual attempting a replication make the same selections based on the information you have provided here?

As explained above to reviewer# 1, the structure of The BMJ papers facilitates identifying the cohort studies by the abstract and title. The BMJ requires that the study design is stated in the title and also the structured abstract includes a study design section. Therefore it was possible to easily identify the cohort studies from the information provided in these sections. Papers that fall in a different category as cohort or longitudinal were labelled as such in these sections. For instance, excluded papers were described as RCTs or surveys in these sections.

Page 9 Line 235-246

For this discussion of the classification of statements of causality, I would point you to research by Adams et al (2017) regarding how such statements are interpreted and classified by participants. Any

such classification system for causality statements might need to be produced by widespread consensus rather than, or in addition to, classifications created by experts. I add this point as a suggestion because the author's final statement on the matter is that there should be "further guidance", but not what that further guidance should be, or how the guidance should be informed.

Thank you for suggesting a relevant paper. We have included a reference to this in the introduction (lines 60-62). Regarding further guidance on the topic, we refer the readers to the guidance developed by Lederer et al. and call for a consensus meeting to help develop a more comprehensive compendium on causal language.

VERSION 2 – REVIEW

REVIEWER	Lex Bouter department of Epidemiology and Data Science Amsterdam University Medical Centers The Netherlands
REVIEW RETURNED	25-Dec-2020
GENERAL COMMENTS	Reviewer comments have been addressed adequately.
REVIEWER	Luke Bratton United Kingdom
REVIEW RETURNED	02-Jan-2021
GENERAL COMMENTS	Thank you for promptly addressing my minor comments and the suggestions of other reviewers. I hope this study can encourage discussion regarding the importance of aligning evidential language with study outcome.